# Cas9/gRNA-Mediated Mutations in *PtrFLA40* and *PtrFLA45* Reveal Redundant Roles in Modulating Wood Cell Size and SCW Synthesis in Poplar

**DOI:** 10.3390/ijms24010427

**Published:** 2022-12-27

**Authors:** Cheng Zhen, Xinguo Hua, Xue Jiang, Guimin Tong, Chunming Li, Chuanping Yang, Yuxiang Cheng

**Affiliations:** State Key Laboratory of Tree Genetics and Breeding, Northeast Forestry University, Harbin 150040, China

**Keywords:** PtrFLAs, Xylem cell size, SCW synthesis, Cas9/gRNA, *Populus trichocarpa*, wood

## Abstract

Fasciclin-like arabinogalactan proteins (FLAs) play an important role in plant development and adaptation to the environment. However, the roles of FLAs in wood formation remain poorly understood. Here, we identified a total of 50 *PtrFLA* genes in poplar. They were classified into four groups: A to D, among which group A was the largest group with 28 members clustered into four branches. Most *PtrFLAs* of group A were dominantly expressed in developing xylem based on microarray and RT-qPCR data. The roles of *PtrFLA40* and *PtrFLA45* in group A were investigated via the Cas9/gRNA-induced mutation lines. Loss of *PtrFLA40* and *PtrFLA45* increased stem length and diameter in *ptrfla40ptrfla45* double mutants, but not in *ptrfla40* or *ptrfla45* single mutants. Further, our findings indicated that the *ptrfla40ptrfla45* mutants enlarged the cell size of xylem fibers and vessels, suggesting a negative modulation in stem xylem cell size. In addition, wood lignin content in the *ptrfla40fla45* mutants was increased by nearly 9%, and the lignin biosynthesis-related genes were significantly up-regulated in the *ptrfla40fla45* mutants, in agreement with the increase in wood lignin content. Overall, Cas9/gRNA-mediated mutations in *PtrFLA40* and *PtrFLA45* reveal redundant roles in modulating wood cell size and secondary cell wall (SCW) synthesis in poplar.

## 1. Introduction

Wood is the most abundant plant biomass on earth and an immense reservoir of fixed carbon. Wood formation undergoes a series of complex developmental processes starting with cambial cell division, followed by xylem cell differentiation and expansion, secondary cell wall (SCW) thickening, and programmed cell death [1,2,3]. In angiosperms, wood is mainly composed of vessel elements and fibers, which are responsible for water conduction and mechanical support, respectively [4]. Wide xylem cells, especially vessels, can improve the transport efficiency by increasing the hydraulic conductivity of trees, but they are also susceptible to dysfunction when experiencing frost and water limitations [5,6,7,8,9]. Notably, cell size is crucial for the proper functioning of xylem vessel elements and fibers in trees. After the xylem cells of the tree are fully expanded, secondary cell wallsstart to deposit inside the primary cell walls. Three major polymers, cellulose, hemicelluloses, and lignin, together constitute the SCWs of xylem cells, which provide support and protection for trees [10]. The structure and chemical components of the SCW produced by plants largely determine the degree of exploitation of wood resources [11]. Therefore, research on the modulation of wood cell size and SCW synthesis in woody plants is helpful for biomass utilization.

Fasciclin-like arabinogalactan proteins (FLAs) belong to a subclass of arabinogalactan proteins (AGPs); aside from AGP motifs, FLAs also have the typical fasciclin-like (FAS) domains. The FAS domain is typically found in animal cell adhesion proteins and is proposed to be important in cell-cell and cell matrix interactions [12]. The control of cell expansion is believed to depend on cell wall structure, interactions between adjacent cells, and wall-membrane interactions [13,14]. Therefore, it is reasonable to infer that plant FLAs may play a role in cell expansion. In response to salt stress, the root tips of *sos5* (loss of *AtFLA4*) mutants have abnormal expansion of epidermal, cortical, and endodermal cells in *Arabidopsis* roots [15], whereas overexpression of *Eucalyptus EgrFLA1* and *EgrFLA3* has enlarged the vessels and fiber/ray cells in transgenic tobacco plants, respectively [16]. In cotton, overexpression of *GhFLA1* leads to an increase in fiber length [17]. It is unknown why different *FLA* genes play opposing roles in plant cell expansion. Thus, whether FLAs have an effect on xylem cell size in trees remains to be identified.

So far, dozens of FLAs have been distinguished in higher plants. According to the number and arrangement of the FAS domains, AGP addition regions, and the presence or absence of the putative glycosyl-phosphatidylinositol (GPI) anchor addition sites, the FLAs are classified into four groups from A to D [18,19,20,21,22]. The FLAs of group A are gradually being revealed to play the roles in SCW biology and to have properties in herbaceous and woody species [16]. *AtFLA11* and *AtFLA12* are exclusively expressed in *Arabidopsis* stems, and the *atfla11/fla12* double mutants reduce stem biomechanics, including tensile strength and elasticity [23]. *Eucalyptus EgrFLA2* and *EgrFLA3* contribute to the SCW properties by decreasing cellulose microfibril angles in xylem fibers and stem flexural strength [24]. Antisense expression of *PtFLA6* in poplar also leads to a reduction in stem flexural strength, accompanied by a change in cell wall composition [25]. In cotton, *GbFLA5*, a crucial SCW-specific protein, contributes to fiber strength by affecting cellulose synthesis and microfibril deposition orientation [26]. The latest findings suggest that *AtFLA11* and *AtFLA12* are possible cell surface sensors regulating SCW development in response to mechanical stimuli [27]. In short, the current report shows changes in stem biomechanics but no change in stem growth phenotypes in group A FLA-transgenic plants.

Poplar has a large biomass accumulation in the terrestrial ecosystem because of its fast growth and high-quality wood. Considering that the poplar FLA gene family evolves many members of group A, the function of these FLAs in stem xylem remains to be widely investigated in trees. In this study, we identified 50 *PtrFLAs* in *Populus trichocarpa* using the well-annotated genome [28]. Expression analysis showed that most of the group A *PtrFLAs* were highly expressed in the developing xylem at the transcription level. Furthermore, we examined the loss-of-function of *PtrFLA40* and *PtrFLA45* in *P. trichocarpa* using the Cas9/gRNA gene editing technique. As a result, *PtrFLA40* and *PtrFLA45* played redundant roles in modulating xylem cell size and SCW synthesis in *P. trichocarpa*.

## 2. Results

### 2.1. Characterization of the FLA Proteins in P. trichocarpa

The amino acid sequences of the reported *Arabidopsis* and *Eucalyptus* FLAs were used as queries to search for the FLAs in the *P. trichocarpa* genome. Further, the candidates were manually examined to determine whether they contained the FAS domains and the AGP-like glycosylated regions with at least two non-contiguous Pros separated by no more than 11 amino acid residues. As a result, the 50 FLAs were identified in *P. trichocarpa* and named PtrFLA1 to PtrFLA50 based on their chromosomal distribution from top to bottom (Appendix A). FLAs are usually secreted proteins and may be attached to the plasma membrane via the C-terminal GPI anchor. As expected, 45 putative PtrFLAs were predicted to contain an *N*-terminal signal peptide by SignalP 5.0, and 32 putative PtrFLAs were predicted to have a GPI anchor sequence by the big-PI predictor. Other information for the *PtrFLA* genes, including gene name, locus, group, protein length (aa), number of putative glycosylated regions, number of FAS-like domains and protein subcellular localization, are shown in Appendix A.

To explore the evolutionary relationship, a phylogenetic tree was constructed with the 50 PtrFLAs, 21 AtFLAs, and 18 EgrFLAs. Similar to the taxonomy of *Arabidopsis* and *Eucalyptus*, PtrFLAs were classified into four phylogenetic groups: A to D (Figure 1). Group A was the largest group with 28 members, PtrFLA2/5/6/8/13/14/17/18/19/20/21/22/23/26/28/29/31/38/39/40/41/42/43/44/45/46/47/48, and, the amino acid similarities among them were 27.78–98.1%; Group B contained five members (PtrFLA10/12/15/30/36), and the amino acid similarities were 70.43–95.06%; Group C contained 10 members (PtrFLA1/3/9/11/16/25/27/33/34/50) with the 20.14–98.89% similarities; and finally the seven members (FLA4/7/24/32/35/37/49) of Group D shared low levels of similarities (17.72–27.31%). Additionally, five pairs of orthologous genes, PtrFLA1/AtFLA14, PtrFLA24/AtFLA21, PtrFLA32/AtFLA20, PtrFLA35/AtFLA19, and PtrFLA27/Eucgr.H00857, were found among *P. trichocarpa*, *Arabidopsis,* and *Eucalyptus*, suggesting that some ancestral FLAs existed prior to the divergence of these species.

Further, it can be seen from the tree branches that the PtrFLAs of group A clustered into four significant sub-clades: 22 PtrFLAs form sub-clade I with AtFLA12; PtrFLA8 and PtrFLA31 form sub-clade II with AtFLA11, Eucgr.B02486 (EgrFLA1), Eucgr.J00938 (EgrFLA2), and Eucgr.A01158 (EgrFLA3); PtrFLA20 and PtrFLA38 with AtFLA6, 9, and 13 form sub-clade III with Eucgr.A01871; PtrFLA5 and PtrFLA26 with AtFLA7 form sub-clade IV with Eucgr.K00711 and Eucgr.A01074 (Figure 1). It indicates that these PtrFLAs are relatively conservative in evolution and may share functions with the corresponding AtFLAs and EgrFLAs. Additionally, 22 PtrFLAs are highly correlated with each other and are homologous with AtFLA12, suggesting that they may have similar functions. However, there are so many evolutionary members of the sub-clade that it is implied that their roles are likely important for the growth and development of *P. trichocarpa*.

### 2.2. Identification of the PtrFLAs Dominantly Expressed in Secondary Xylem

To identify the *PtrFLAs* involved in secondary xylem development, we collected their transcript abundances in diverse tissues and organs, including young leaf, mature leaf, roots, xylem, female catkins, and male catkins, from microarray data of the *Populus* eFP browser. As a result, the 34 *PtrFLA* genes were found in the microarray data (Appendix A), and the heat map demonstrated that most *PtrFLAs* had preferential or tissue-specific expression profiles (Appendix A). *PtrFLA1* and *PtrFLA11* were highly and preferentially expressed in catkins. Apparently, the 16 *PtrFLAs* from group A, *PtrFLA2/5/6/13/17/18/21/22/23/26/28/29/31/40/42/45*, were dominantly expressed in developing xylem. The findings suggest that the *PtrFLAs* of group A tend to play a role in secondary xylem development.

As group A of the FLA family was the largest group with 28 members in *P. trichocarpa* (Figure 1; Appendix A), we further examined their expression levels by RT-qPCR in developing xylem, phloem, leaf, and shoot (Figure 2). The expression profiles of most *PtrFLAs* were consistent with those from the microarray data. Except for *PtrFLA38/44/46*, other *PtrFLAs* of group A were dominantly expressed in developing xylem. As the developing shoot provides a gradual gradient from primary to secondary growth that facilitates the identification of genes in wood formation [29], we analyzed the expression of group A *PtrFLAs* in the stem internodes with different lignification levels (Appendix A). These *PtrFLAs* were hardly expressed in the IN1 and IN2 tissues undergoing the primary growth stage, and conversely, they were highly expressed in the lignified stem INs. Overall, these results suggest that the *PtrFLAs* of group A should have the most important roles in the wood formation of *P. trichocarpa*.

### 2.3. Production of Cas9/gRNA-Mediated ptrfla40, ptrfla45, and ptrfla40 ptrfla45 Mutants

To explore the role of the *PtrFLAs* of group A in wood formation, *PtrFLA40* and *PtrFLA45* were selected for the test, considering that they are dominantly expressed in developing xylem and their amino acid identities are up to 95.2% (Figure 1 and Figure 2). We generated single-gene mutants (*ptrfla40* and *ptrfla45*) and double mutants (*ptrfla40ptrfla45*) using the Cas9/gRNA technique. Two gRNAs were designed for each gene to generate multiple mutants (Figure 3A,D). After transforming *P. trichocarpa*, a total of 6, 7, and 9 independent transgenic lines were obtained for editing *PtrFLA40*, *PtrFLA45,* and *PtrFLA40/45*, respectively. Detection of edited target sites by DNA sequencing showed a nucleotide insertion in partial lines and small deletions of the nucleotides ranging from one to several in a majority of lines (Appendix A). The insertions and deletions brought about frameshift mutations in protein-coding sequences, leading to the premature generation of termination codons (Figure 3C,F). Among these, the lines 2# and 3#, 1# and 2#, and 2# and 5# showed homozygous mutations in *PtrFLA40*, *PtrFLA45,* and *PtrFLA40/45*, respectively (Figure 3B,E), which were used for gene function analyses.

### 2.4. Loss of PtrFLA40 and PtrFLA45 Increases Stem Length and Diameter in Double Mutants

To understand the role of *PtrFLA40* and/or *PtrFLA45* in wood formation, we examined the phenotypes of the *ptrfla40*, *ptrfla45*, and *ptrfla40fla45* mutants grown in the greenhouse. Compared with the wild-type, the *ptrfla40* and *ptrfla45* single mutants displayed no observable difference in height growth, whereas the *ptrfla40fla45* double mutants were definitely tall in plant height (Figure 4A). The statistical data further showed that the plant heights of the *ptrfla40fla45* double mutants were taller than those of the wild-type and single mutants (Figure 4B). To detect whether growth and development are improved in the mutants, we counted the internode numbers of 4-month-old wild-type and mutant plants in the greenhouse. No significant difference in internode number was observed between wild-type and *ptrfla40*, *ptrfla45*, and *ptrfla40fla45* mutants (Figure 4C), suggesting that an increase in double mutant height growth was probably caused by the increase in internode elongation. The further measured data indicated that the length of the 7th, 9th, 11th, 13th, and 15th internodes in the *ptrfla40fla45* double mutants all increased in comparison with those of the wild-type and single mutants (Figure 4D), which was consistent with the data that the plant heights of the *ptrfla40fla45* double mutants were taller than those of the wild-type and single mutants. In addition, the *ptrfla40fla45* double mutants had a slight but significant increment in stem diameter compared with wild-type, while the *ptrfla40* and *ptrfla45* single mutants did not display obvious phenotypic alterations (Figure 4E). Taken together, these results suggest that *PtrFLA40* and *PtrFLA45* play a redundant role in stem length and diameter in *P. trichocarpa*.

### 2.5. PtrFLA40 and PtrFLA45 Negatively Modulate Stem Xylem Cell Size in P. trichocarpa

To investigate the role of *PtrFLA40* and *PtrFLA45* in secondary xylem development, we observed the cross-sections of the stems from 4-month-old wild-type plants and mutants. The radial width of bark and pith of *ptrfla40, ptrfla45,* and *ptrfla40fla45* double mutants did not differ significantly from the wild-type. However, the width of secondary xylem in the stems of the *ptrfla40fla45* double mutants was significantly increased (Figure 5A,B). Further statistical data showed that the xylem layers of the *ptrfla40fla45* double mutants were greater than those of wild-type and the *ptrfla40* and *ptrfla45* single mutants (Figure 5C). 

In addition, a microscope photograph showed that the pore diameter of xylem fibers and vessels in the *ptrfla40fla45* double mutants was bigger than that of wild-type, *ptrfla40,* and *ptrfla45* plants (Figure 5A). We further measured the change in xylem cell size in the *ptrfla40fla45* double mutants. Consistent with the conjecture, the statistical data showed that the proportion of xylem fibers and vessels with a larger pore diameter in the *ptrfla40fla45* double mutants was significantly higher than that of wild-type plants, but there was no difference between the wild-type and the two single mutants (Figure 6A,B). Furthermore, we examined the longitudinal measurement of the xylem fibers and vessels in wild-type and mutant plants. No significant differences in the length of xylem fibers and vessels were observed between the wild-type and single mutants. More strikingly, the number of fibers shorter than 600 μm in the *ptrfla40fla45* double mutants was significantly less than that in the wild-type plants (Figure 6C). Conversely, the number of fibers with a length range of 600 to 800 μm accounted for more than 42% in the *ptrfla40fla45* double mutants, whereas that of wild-type plants merely reached about 25% (Figure 6C). Simultaneously, there was about a 1.2% proportion of fibers longer than 800μm in the *ptrfla40fla45* double mutants but almost none in the wild-type and *ptrfla40* and *ptrfla45* single mutants (Figure 6C). Similar differences were found in the length of vessels in the *ptrfla40fla45* double mutants (Figure 6D). Overall, our findings indicate that loss of *PtrFLA40* and *PtrFLA45* in *P. trichocarpa* increases the cell sizes of xylem fibers and vessels, implying a negative modulation of stem xylem cell size.

### 2.6. Deletion of PtrFLA40 and PtrFLA45 alters Wood SCW Synthesis in ptrfla40fla45 Mutants

To test whether the mutation of *PtrFLA40* and *PtrFLA45* affects the synthesis of SCW, we examined the contents of lignin, cellulose, and hemicellulose in wild-type and mutant wood. The data showed that no significant difference in lignin or cellulose content was shown between single mutants and wild-type plants. In contrast, compared with wild-type, wood lignin content in the *ptrfla40fla45* mutants was increased by nearly 9%, while wood cellulose content was slightly decreased (Table 1). In addition, the contents of xylose, the main monosaccharide of hemicellulose, and the low concentration of monosaccharides (glucose, mannose, galactose, arabinose, rhamnose, and fucose) in these mutants were not significantly different from those in wild-type plants, indicating that accumulation of wood hemicellulose did not change (Table 1). Besides, the wall thickness of the secondary xylem fibers showed no significant differences between wild-type and these mutants (Appendix A).

Further, expression of SCW-related genes was examined by RT-qPCR in *ptrfla40*, *ptrfla45*, and *ptrfla40fla45* mutants. The lignin biosynthesis enzyme genes, *PtrPAL4*, *Ptr4CL3*, *PtrC3H3*, *PtrCcoAOMT2*, *PtrF5H2,* and *PtrCOMT2*, were significantly up-regulated in the *ptrfla40fla45* mutants (Figure 7), which was consistent with the increase in wood lignin content. In *ptrfla40* and *ptrfla45* single mutants, only the *PtrCOMT2* gene was moderately up-regulated at the transcription level (Figure 7). These findings indicate that deletion of *PtrFLA40* and *PtrFLA45* affects SCW synthesis, mainly in lignin synthesis-related genes. In view of the fact that the xylem vessels and fibers of the *ptrfla40fla45* mutants were larger than those of the wild-type, we further studied the expression of expansins, which were largely expressed in poplar stems. The results showed that there was no significant difference in the expression levels of these expansin genes and *PtrCYCD3;3* between wild-type and these mutants (Appendix A).

## 3. Discussion

In the present study, we have identified 50 FLA genes in poplar, all of which contain one or two FAS domains. According to genome information of various plant species, the members of the FLAs significantly increase from lower plants to higher plants, and the 4 or 6 FAS domains of the FLA protein decrease to one or two from red or green algae to land plants during evolution [30]. It is speculated that the expansion of the FLA genes and the reduced number of FAS domains during evolution may likely adapt to the diversified roles in higher plant growth and development. In *Arabidopsis*, group A FLAs with one FAS domain contain six members [22]. However, the number of group A FLAs in *P. trichocarpa* was up to 28, most of which were highly expressed in developing xylem, as suggested by the microarray and RT-qPCR data (Figure 2 and Appendix A). The findings hint that they have diversified roles in stem growth and development in poplar.

Our findings have shown that *PtrFLA40* and *PtrFLA45* negatively modulate the size of xylem cells in poplar stems. In the phylogenetic tree, 22 PtrFLAs, including the two members, clustered with *Arabidopsis* AtFLA12 into sub-clade I of group A FLAs (Figure 1)*. AtFLA12* contributes to stem strength, whereas *AtFLA12*-overexpression plants and *atfla12* mutants have inflorescence stem cells that display similar size and shape as wild-type cells [23,27]. In the *ptrfla40fla45* mutants, loss of *PtrFLA40* and *PtrFLA45* significantly enlarged cell dimensions of xylem fibers and vessels (Figure 6), indicating involvement in xylem development. Recently, compared with the controls, overexpression of *EucalyptusEgrFLA1* and *EgrFLA3* led to larger vessels and fiber/ray cells in transgenic tobacco plants, respectively [16]. The findings indicate that *EgrFLA1* and *EgrFLA3* positively modulate the size of vessels and fiber/ray cells, which is distinct from the modulation of *PtrFLA40*/*45* in the size of xylem cells. It is likely attributed to the fact that *EgrFLA1/3* and *PtrFLA40/45* belong to different sub-clades of group A FLAs (Figure 1). In addition, an earlier study reported that *AtFLA4* is strongly expressed in root tips, and loss-of-function plants exhibit abnormal expansion of epidermis and cortex cells in root tips [15]. AtFLA4 is a member of group C FLAs, in which each has two FAS domains, different from PtrFLA40/45 with one FAS domain.

It is believed that the plant cell wall is a complex network with potential loosening and expansion sites [31,32,33]. Expansins are the main cell wall loosening agents, which promote the flexibility of plant cell wall by loosening the hydrogen bond between cellulose microfibrils and matrix polymers [34]. We examined the transcription levels of five expansin genes in the *ptrfla40fla45* mutants to explore modulation of *PtrFLA40/45* in xylem cell dimensions at molecular levels. These expansin genes have shown no significant difference in the wild-type and mutants at the transcriptional level (Appendix A), suggesting that *PtrFLA40/45* do not regulate or mediate the expansin genes to alter xylem cell size. A proposed model indicates that the FAS domains can interact with each other, which can generate an interlacing network that inhibits cell elasticity [35]. Therefore, we infer that knockout of *PtrFLA40*/*45* may promote the relaxation of xylem cell walls through the scission of a stress-bearing crosslink, leading to the enlargement of xylem cell dimensions. Certainly, we cannot exclude the possibility that deletion of *PtrFLA40/45* in poplar may increase nonautonomous cell expansion in multiple directions.

Phenotypic observations indicate that *PtrFLA40* and *PtrFLA45* play a negative role in poplar height. However, the *ptrfla40fla45* mutants do not quicken growth and development, as suggested by the same stem internode number as the wild-type and mutants (Figure 4C). It is the reason that an increase in each internode length contributes to the mutant height growth. Apparently, the mutant stem internode elongation is ascribed to the enlargement of cell size in xylem fibers and vessels. Besides, the *ptrfla40fla45* mutants have more xylem cell layers than the wild-type and single mutants (Figure 5B,C). Secondary xylem derives from the periclinal divisions of vascular cambium, but *PtrCYCD3;3*, a marker gene of cambial activity [36,37], displays similar transcription levels in wild-type, single, and double mutants (Appendix A). This suggests no significant change in cambium activity in the *ptrfla40fla45* mutants. It is speculated that cell enlargement may shorten the time for xylem cell maturity, thus accumulating more xylem cell layers in the *ptrfla40fla45* mutants.

Lignin is an important component of wood SCWs. In addition to xylem development, *PtrFLA40/45* modulate SCW synthesis in wood, as suggested by a significant increase in lignin content in the *ptrfla40fla45* mutants (Table 1). A most recent study has considered that *AtFLA11* and *AtFLA12* are possible cell surface sensors regulating SCW development under mechanical stimuli [27]. Compared with wild-type, the *fla12* mutants have increased lignin content and decreased cellulose content [23], which is the same as the *ptrfla40fla45* mutants. Considering the same sub-clade as AtFLA12, we propose that PtrFLA40/45 may share a similar role in xylem SCW synthesis. Our findings have further shown that six lignin biosynthesis enzyme genes, especially *PtrF5H2* and *PtrCOMT2*, are significantly up-regulated in the *ptrfla40fla45* mutants (Figure 7). F5H is a limiting enzyme in the biosynthesis of S-monolignol in angiosperms, and overexpression of the *F5H* gene can increase the content of S units [38,39]. COMT plays the main role in the methylation of 5-hydroxyconiferaldehyde, a precursor of lignin S units, and its RNAi suppression has reduced 30% of the S units of the lignin in barley [40]. This suggests that loss of *PtrFLA40/45* may mainly increase the synthesis of S-lignin units. However, more evidence is required to verify the proposal. In addition, a dozen FLA genes are highly expressed during tension wood (TW) formation in poplar, and some of them have been regulated by GA signaling to participate in TW formation [41,42,43]. This provides a possible reason why many *PtrFLAs* of group A are more specifically expressed in poplar xylem.

## 4. Materials and Methods

### 4.1. Plant Material and Growth Conditions

The *P. trichocarpa* (Nisqually-1) used in this study was cultured in the greenhouse of the Northeast Forestry University, China. Wild-type and mutant plantlets were propagated in vitro on Lloyd & McCown Woody Plant Basal Medium w/Vitamins (WPM; PhytoTech Lab, L449) supplemented with 2.5% (*w*/*v*) sucrose. Young trees grew in the greenhouse under a 16 h photoperiod and approximately 200 µmol m^−2^ s^−1^ at 23–25 °C. The 4-month-old wild-type trees were used to detect gene expression levels in different tissues, including developing xylem, phloem, leaf, and shoot. The 1st–5th, 7th, 9th, 11th, and 13th internodes were collected as described in a previous study [44]. In addition, 4-month-old wild-type and mutant trees were used for phenotype analysis.

### 4.2. Identification of P. trichocarpa FLAs (PtrFLAs)

The protein sequences of 21 Arabidopsis FLAs [22] and 18 Eucalyptus FLAs [16] were used as queries to search the *P. trichocarpa* genome on Phytozome v13 (https://phytozome.jgi.doe.gov/pz/portal.html (accessed on 23 February 2021)) [45] to identify potential PtrFLAs. The repetitive genes were removed, and all potential PtrFLAs were scanned with the SMART database (http://smart.embl-heidelberg.de/ (accessed on 26 March 2021)) [46] to identify the FAS domain (SM00554). Subsequently, all objective FLAs were manually analyzed for the presence of putative AGP addition regions which contain at least two non-contiguous Pro residues, for example, (A/S/T) P(A/S/T) P separated by no more than 11 amino acid residues [22]. Ultimately, *N*-terminal signal peptide regions and GPI-anchor addition sites were searched using SignalP 5.0 (http://www.cbs.dtu.dk/services/SignalP/ (accessed on 28 March 2021)) [47] and the big-PI Predictor (http://mendel.imp.ac.at/sat/gpi/gpi_server.html (accessed on 28 March 2021)) [48], respectively. The signal peptide sequence of each protein was removed, and multiple sequence alignment was carried out by Clustal X 2.0 [49]. The phylogenetic tree was constructed with MEGA 6.0 using the neighbor-joining (NJ) method with 1000 bootstrap replicates [50].

### 4.3. Microarray Data, RNA Extraction, RT-qPCR and RT-PCR Analyses

Tissue-specific expression data on the *PtrFLAs* were downloaded from the Populus eFP browser (http://bar.utoronto.ca/efppop/cgi-bin/efpWeb.cgi (accessed on 6 May 2021)). The heat map was generated by Heat Map Illustrator (HemI) with the default settings. Total RNA was extracted using plant RNA extraction reagents (Bio-Flux, Beijing, China). RNA concentration and quality were measured in a Nanodrop ND-1000 (Thermo Scientific, Waltham, MA, USA) and cDNA was synthesized using the Prime Script RT reagent Kit (TaKaRa, Dalian, China) according to the manufacturer’s instructions. Quantitative real-time PCR (RT-qPCR) analysis was performed in the ABI Prism 7500 real-time PCR system (Applied Biosystems, Waltham, MA, USA) using the SYBR Green (TaKaRa, Dalian, China). The *PtActin2* gene was used as the reference, and the 2^−∆∆Ct^ method was used to calculate gene expression levels. In addition, expression of the *PtrFLAs* in different stem internodes was examined by RT-PCR analysis. The reaction mixture (20 μL) consisted of 2 μL 10× Taq buffer, 2 μL of dNTP mixture, 1 μL of Taq DNA polymerase, 1 μL of the cDNA template, 0.5 μL of each gene-specific primer (10 μM), and 13 μL of distilled H_2_O. The PCR parameters were 94 °C for 5 min, followed by 28–35 cycles of 94 °C for 30 s, 58 °C for 30s, 72 °C for 30 s, and 72 °C for 7 min. Expression of the *PtActin2* gene was used as the control, and PCR products were detected on the agarose gels. All primers are listed in Appendix A.

### 4.4. gRNA Design and Vector Construction

For the mutations in *PtrFLA40* and/or *PtrFLA45*, we used the Cas9/gRNA gene editing technique. Two gRNAs for each gene were manually selected, and a BLASTN search was carried out on each target site in the *P. trichocarpa* genome to ensure target specificity. The candidate targets were close to the 5′-ends of the coding sequence (CDS), which are apt to produce translation terminators early in the mutated gene edited by Cas9/gRNA. For construction of the Cas9/gRNA plant transformation binary vector, we used the method described by Xing [51]. In brief, the PCR fragment was amplified using pCBC-DT1T2 as the template, and the purified PCR fragment and pHSE401 binary vector were used to set up restriction-ligation reactions using BsaI and T4 Ligase. The reaction was incubated for 5 h at 37 °C, 5 min at 50 °C and 10 min at 80 °C. Finally, the sequenced pHSE401-gRNA vector was transferred into Agrobacteria strain GV3101. Detailed information, including vector construction primers and sequencing primers, can be found in Appendix A.

### 4.5. Genetic Transformation and Detection of Cas9/gRNA-Induced Mutations

Agrobacterium-mediated transformation of Nisqually-1 was carried out according to our transformation protocol [52]. Hygromycin was used as the resistance screening marker in this report. Stems of 4-week-old sterile plantlets were cut into small explants and inoculated in suspension with Agrobacterium (OD_600_ = 0.5) for 20 min. Explants were co-cultivated with Agrobacterium for 2 days in the dark and subsequently distributed on selection medium containing 10 mg L^−1^ of hygromycin. After 25 days, the explants were transferred to fresh selection medium containing 5 mg L^−1^ of hygromycin, and the resistant shoots could be induced after another 15 days. The hygromycin-resistant shoots were cut and placed on medium with 5 mg L^−1^ hygromycin for rooting.

For detection of the Cas9/gRNA-induced mutations, genomic DNAs were extracted from the leaves of wild-type and transgenic plantlets using the Plant Genomic DNA Extraction Kit (Bioteke, Wuxi, China). The genomic DNA fragments flanking the gRNA target sites of the edited gene were cloned by PCR amplification with gene-specific primers. The amplicons were purified and cloned into the pMD18-T vector (Takara, Beijing, China), and at least 20 positive clones for each line were sequenced to detect the edited mutations. All primers are shown in Appendix A.

### 4.6. Analyses of Growth, Stem Morphology and Xylem Cell Size

Wild-type and mutant plants were planted into the soil in the greenhouse, and the height of the plants was measured after four months of growth. When calculating the number of internodes, the first stem internode (IN1) was defined as being above the first expanded leaf at the apex. The length and diameter of internodes 5th, 7th, 9th, 11th, 13th, and 15th were measured with a ruler and a vernier caliper, respectively. For stem morphology analysis, stem segments of the 4-month-old plants were gradually fixed, embedded, sectioned, and stained with toluidine blue, as described by Liu [29]. The radial width of bark, xylem, and pith were measured, and the number of cambial, xylem, and phloem cell layers was counted in the transverse sections. For xylem cell size analysis, the length and width of fibers and vessels in the 20th internode of 4-month-old wild-type and mutants were analyzed using the method from Lautner et al. [53]. In brief, the peeled stems were cut into small pieces with a razor blade and incubated in a maceration solution (10% HNO_3_ and 10% CrO_3_ in *v*/*v*, 1:1) for 2–4 h at 60 °C. After small pieces of the stems were washed with distilled water, the fibers and vessels were separated from each other by shaking in water, stained with 0.1% acid magenta, and imaged under a light microscope (BX43, Olympus, Shinjuku, Japan). The length and width of fiber and vessel cells were measured by ImageJ software. On average, 500 fibers and 200 vessels per sample were measured, and three biological replications were performed.

### 4.7. Scanning Electron Microscopy (SEM) and Wood Composition Assay

For scanning electron microscopy (SEM), free-hand cross-sections of fresh 12th stem internodes of 4-month-old wild-type and mutant cells were coated with gold (Au) at 10 mA for 50 s. The gold-coated samples were transferred to the benchtop SEM chamber (JCM-5000, JEOL, Tokyo, Japan) and imaged for analyzing wall thickness of mature xylem fibers. Three individual plants from each line were used for analysis.

For the wood composition assay described in our previous study [54], the basal stems from 4-month-old wild-type and mutant trees were peeled, dried at 55 °C, and ground into powder. Then, 0.5 g sample powder was washed successively with 70% ethanol aqueous, chloroform/methanol (1:1 *v*/*v*) solution, and acetone, and the insoluble residues were air-dried as wood cell wall materials for lignin and cellulose content assays. Lignin content was assayed using the acetyl bromide spectrophotometric method. Hemicellulose content was determined by the GC-MS method, as described previously [55]. For each composition analysis, cell wall materials from three tree wood species were mixed for an assay, and three biological repeats were performed for each mutation line.

### 4.8. Statistical Analysis

All statistical tests and data analyses were performed using SPSS version 24.0. Values are the means ± standard deviation (SD), and asterisks indicate statistical significance at different levels (* *p* < 0.05, ** *p* < 0.01).

## Figures and Tables

**Figure 1 ijms-24-00427-f001:**
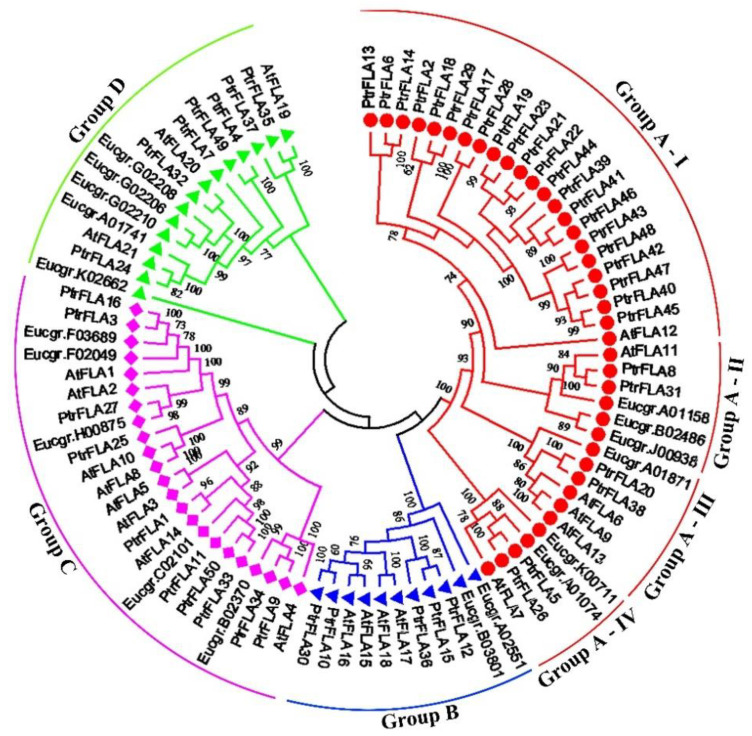
Phylogenetic analysis of *P. trichocarpa*, *Arabidopsis,* and *Eucalyptus* FLAs. 50 PtrFLAs, 21 AtFLAs, and 18 EgrFLAs were aligned with Clustal X 2.0, and the phylogenetic tree was constructed using the neighbor-joining method with 1000 bootstrap replication by MEGA 6.0. All FLAs were classified into four groups: A, B, C, and D, respectively. Group A FLAs showed four sub-clades: I, II, III, and IV in the phylogenetic tree.

**Figure 2 ijms-24-00427-f002:**
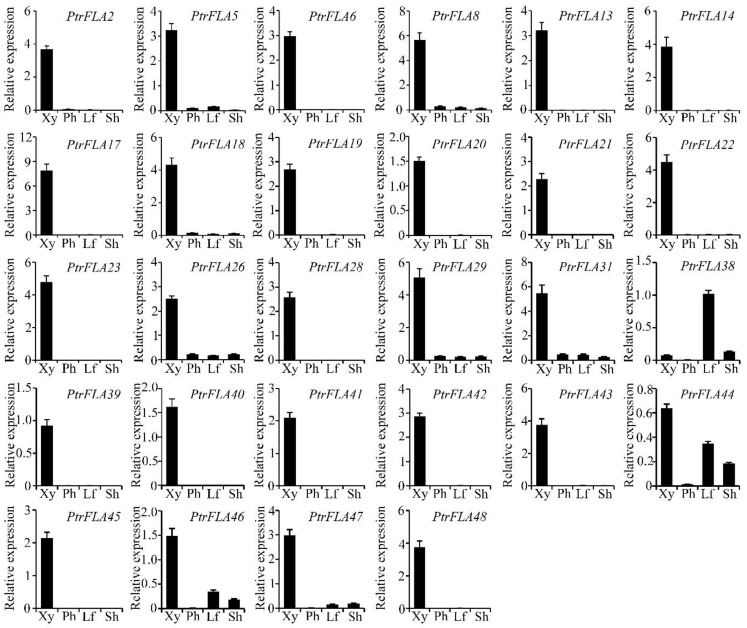
Expression profile of the group A FLAs in *P. trichocarpa* using RT-qPCR. Different tissues included developing xylem, phloem, leaf, and shoot. The expression of *PtActin2* was used as an internal control. Xy, xylem; Ph, phloem; Lf, leaf; Sh, shoot.

**Figure 3 ijms-24-00427-f003:**
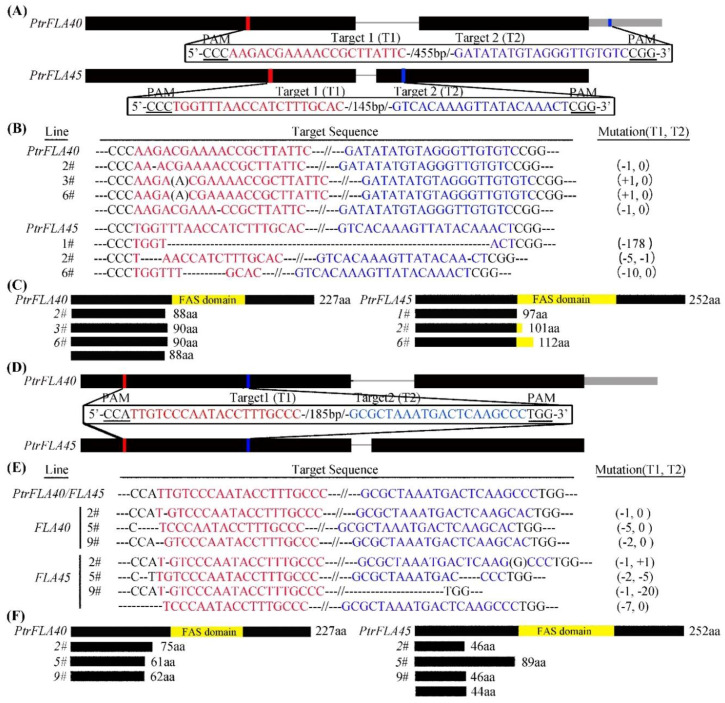
Cas9/gRNA-mediated mutations in the *PtrFLA40*, *PtrFLA45,* and *PtrFLA40/45* genes. (**A**,**D**) Two different target sites in red or blue were selected for editing *PtrFLA40*, *PtrFLA45*, or *PtrFLA40/45*, respectively. Black and gray boxes represent the exons and 3′UTR of the gene, respectively. (**B**,**E**) The mutations in *PtrFLA40*, *PtrFLA45*, or *PtrFLA40/45* in the mutants (*ptrfla40-2#, -3#* and *-6#*, *ptrfla45-1#, -2#,* and *-6#*, *ptrfla40fla45-2#, -5#,* and *-9#*). Black letters in parentheses and dashes represent the insertion and deletion nucleotides, and the number is indicated at the right side of each panel. (**C**,**F**) Deduced amino acids of protein-coding regions from the Cas9/gRNA-edited genes in *ptrfla40*, *ptrfla45,* and *ptrfla40fla45* mutants. The amino acid number of the mutated proteins is indicated at the right. The yellow boxes represent the FAS domain.

**Figure 4 ijms-24-00427-f004:**
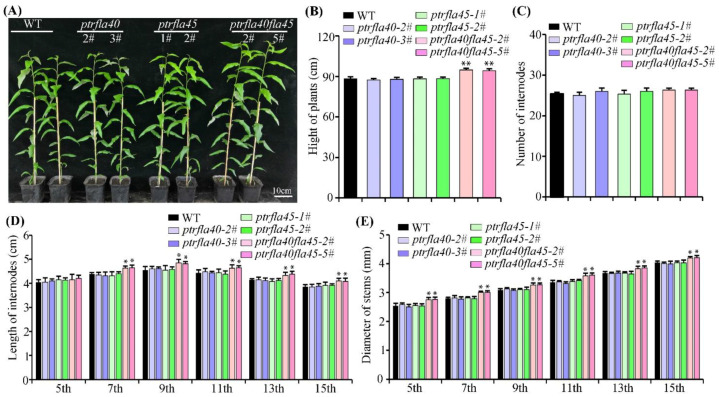
Phenotypes of the *ptrfla40*, *ptrfla45,* and *ptrfla40fla45* mutants. (**A**) Morphology of wild-type (WT) and mutants grown for 4 months in the greenhouse. (**B**,**C**) Plant height and internode number of wild-type, *ptrfla40, ptrfla45*, and *ptrfla40fla45* mutants. (**D**,**E**) Length and diameter of the 5th, 7th, 9th, 11th, 13th, and 15th stem internodes (starting from the morphological apex) in the WT and mutants. Data are means ± SD (*n* = 3). Bar: 10cm. Student’s *t*-test: *, *p* < 0.05; **, *p* < 0.01.

**Figure 5 ijms-24-00427-f005:**
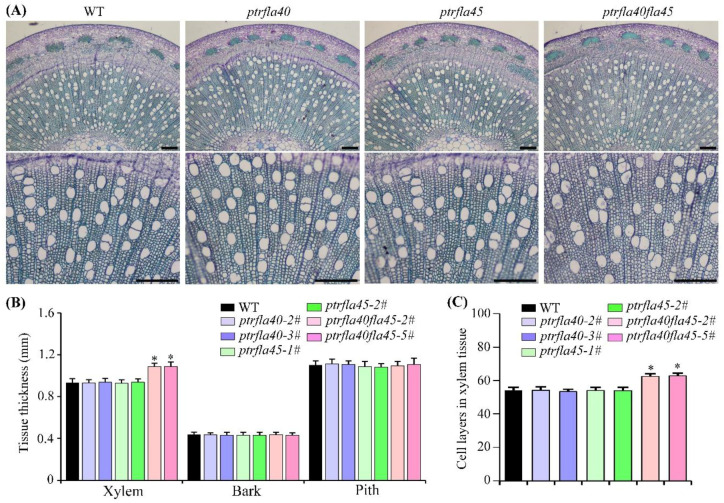
Microscopical observations in the stem tissues of wild-type, *ptrfla40*, *ptrfla45,* and *ptrfla40fla45* mutants. (**A**) Cross-sections of the 12th internode from 4-month-old representative wild-type (WT) plants and mutants. (**B**) Statistical analysis of the radial width of pith, xylem, and bark tissues. (**C**) Layers of xylem cells. Data are means ± SD (*n* = 3). Bars: 200 μm. Student’s *t*-test: *, *p* < 0.05.

**Figure 6 ijms-24-00427-f006:**
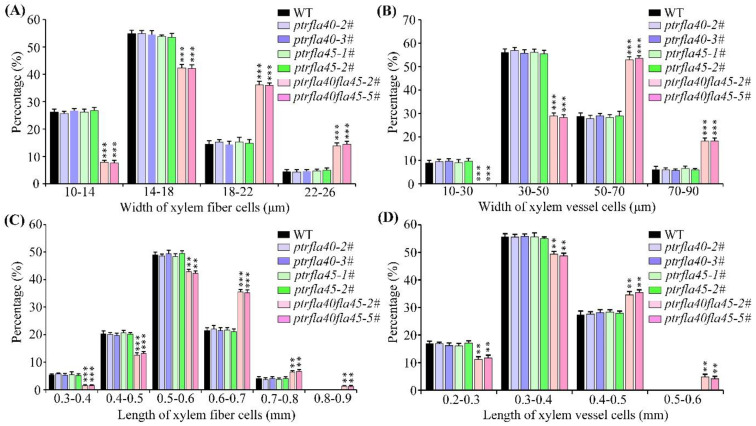
Statistical analysis of the width and length of xylem fiber and vessel cells in wild-type (WT), ptrfla40, ptrfla45, and ptrfla40fla45 mutants. (**A**,**B**) Transverse size percentages of the different subgroups of WT and mutant xylem fibers and vessels, respectively. (**C**,**D**) Percentage of the different subgroups of WT and mutant xylem fibers and vessels in longitudinal size, respectively. Student’s *t*-test: **, *p* < 0.01; ***, *p* < 0.001.

**Figure 7 ijms-24-00427-f007:**
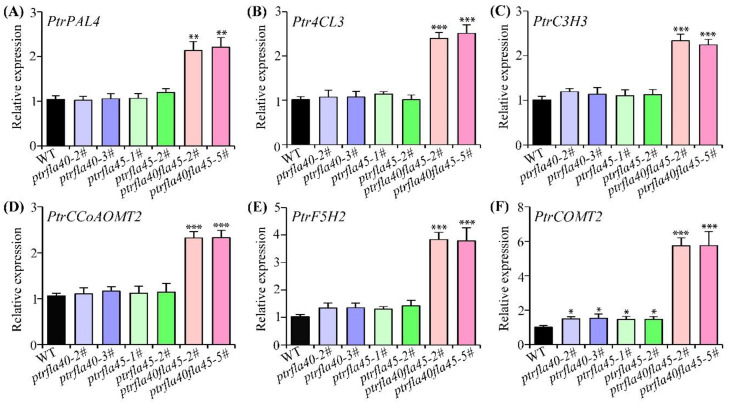
Transcription levels of lignin biosynthesis enzyme genes in wild-type (WT), *ptrfla40, ptrfla45,* and *ptrfla40fla45* mutants by RT-qPCR analysis. (**A**–**F**) *PtrPAL4*, *Ptr4CL3*, *PtrC3H3*, *PtrCCoAOMT2*, *PtrF5H2* and *PtrCOMT2* were quantitatively detected in stems of WT, *ptrfla40, ptrfla45,* and *ptrfla40fla45* mutants. *PtActin2* was used as an internal control. PAL, phenylalanine ammonia-lyase; 4CL, 4-coumarate: CoA ligase; C3H, *p*-coumarate 3-hydroxylase; CCoAOMT, caffeoyl-CoA O-methyltransferase; F5H, ferulate 5-hydroxylase; COMT, caffeic acid O-methyltransferase. Data are means ± SD (*n* = 3). Student’s *t*-test: *, *p* < 0.05; **, *p* < 0.01; ***, *p* < 0.001.

**Table 1 ijms-24-00427-t001:** Wood composition of wild-type and mutant (*ptrfla40, ptrfla45,* and *ptrfla40fla45*) plants.

Wood Composition	Wild-Type	*ptrfla40-2#*	*ptrfla40-3#*	*ptrfla45-1#*	*ptrfla45-2#*	*ptrfla40fla45-2#*	*ptrfla40fla45-5#*
Cellulose (%)	41.67 ± 0.42	41.43 ± 0.37	41.58 ± 0.35	41.73 ± 0.56	41.47 ± 0.52	39.62 ± 0.21	39.42 ± 0.33
Lignin (%)	21.89 ± 0.35	22.35 ± 0.37	21.87 ± 0.53	22.26 ± 0.53	22.41 ± 0.28	25.31 ± 0.42 **	24.89 ± 0.62 **
Polysaccharide ^a^							
Xylose	201.46 ± 4.58	198.48 ± 5.13	200.87 ± 4.59	203.59 ± 5.18	200.52 ± 4.13	197.12 ± 2.34	199.14 ± 3.39
Glucose	55.65 ± 1.21	55.12 ± 0.56	55.83 ± 0.99	54.79 ± 1.32	55.39 ± 0.86	56.21 ± 0.31	55.08 ± 0.69
Mannose	14.28 ± 0.33	14.15 ± 0.21	14.19 ± 0.09	14.06 ± 0.31	14.13 ± 0.22	14.34 ± 0.18	14.41 ± 0.12
Galactose	8.46 ± 0.32	8.38 ± 0.21	8.45 ± 0.15	8.49 ± 0.24	8.35 ± 0.32	8.54 ± 0.22	8.36 ± 0.34
Arabinose	4.25 ± 0.18	4.15 ± 0.29	4.21 ± 0.17	4.23 ± 0.16	4.31 ± 0.09	4.37 ± 0.07	4.27 ± 0.11
Rhamnose	4.41 ± 0.23	4.38 ± 0.17	4.23 ± 0.33	4.09 ± 0.25	4.59 ± 0.31	4.43 ± 0.28	4.68 ± 0.35
Fucose	0.99 ± 0.16	1.02 ± 0.12	1.02 ± 0.05	0.98 ± 0.21	1.02 ± 0.09	0.99 ± 0.06	1.01 ± 0.14

Four-month-old plants were used for the tests. Data are means ± SD (*n* = 3). Student’s *t* test: **, *p* < 0.01. ^a^ Data are shown as µg mg^−1^ cell wall residues.

## Data Availability

Not applicable.

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
