# Peer review of "Cas9/gRNA-Mediated Mutations in PtrFLA40 and PtrFLA45 Reveal Redundant Roles in Modulating Wood Cell Size and SCW Synthesis in Poplar"

_ijms, 2022, doi:10.3390/ijms24010427_

Round 1
Reviewer 1 Report
The manuscript is very interesting but minor revision required as bellow.
Line-78. The gene name should be in italic.
Line 83. “FLAs” should be write as “ FLA proteins” in sub head.
PtrFLA may be write as “PtFLA”. please check through out manuscript.
Line-98, 111, 112. The gene name should be in italic. Arabidopsis should also be in italic. I noticed that at many places, the gene name is not in italic. Please correct it through out manuscript.
Fig-2. The statistical analysis is missing. Please perform the significance test and put it in the graphs in the form of letters or *.
Author Response
Line 78. The gene name should be in italic.
Answer: Thanks. We have corrected “PtrFLAs” into italic in line 79.
Line 83. “FLAs” should be writed as “FLA proteins” in sub head.
Answer: Thanks. We have revised it in line 84.
PtrFLA may be writed as “PtFLA”. Please check throughout manuscript.
Answer: Thanks for your helpful suggestion. You are right! To avoid confusion with other poplar species (for instance, Populus tomentosa), we have abbreviated Populus trichocarpa as Ptr, which also refers to some literatures.
Line-98, 111, 112. The gene name should be in italic. Arabidopsis should also be in italic. I noticed that at many places, the gene name is not in italic. Please correct it throughout manuscript.
Answer: Thanks. We have corrected “Arabidopsis” into italic in lines 52, 85, 99, 108, 112, 294 and 301. We have corrected “Eucalyptus” into italic in lines 85, 109 and 112. The words “PtrFLAs”, “AtFLAs” and “EgrFLAs” in lines 99, 112 and 113 (original lines 98, 111 and 112) represent proteins, and we have not corrected them into italics. In addition, we have made the following changes in other places that need to be corrected into italics, for instance, “PtrFLAs” into italic in lines 130, 134, 136, 138, 150, 361, 475 and 476.
Fig-2. The statistical analysis is missing. Please perform the significance test and put it in the graphs in the form of letters or *.
Answer: Thank you for your helpful suggestion. Figure 2 shows the relative transcriptional abundance of group A PtrFLAs in diverse tissues. As we did not set certain tissue as the control, the significance test seems to be inessential. If essential, we perform it.
Reviewer 2 Report
The manuscript "Cas9/gRNA‐mediated mutations in PtrFLA40 and PtrFLA45 reveal redundant roles in modulating wood cell size and SCW synthesis in poplar" is written in good scientific language. The Fasciclin‐like arabinogalactan proteins play an essential role in plant development and environmental adaptation, and their roles in wood formation remain to be widely investigated. The scientific soundness and interest to the readers are high. Modern and relevant analyzes are applied. The obtained results are processed and interpreted adequately. Тhe discussion part is detailed and follows the results obtained.
Comments:
Line 37 - Abbreviation is not necessary!
Line 64 - Instead of EgrFLA2 and 3, it is better to write EgrFLA2 and EgrFLA3.
Line 85 - Which one is DB?
Figure 2 appears before the citation in the text!
Figures 2, and 6, part of Figures 4, 5, and 7, are poor quality.
Author Response
Line 37 - Abbreviation is not necessary!
Answer: Thanks. We have revised it.
Line 64 - Instead of EgrFLA2 and 3, it is better to write EgrFLA2 and EgrFLA3.
Answer: Thanks. We have revised it in line 64.
Line 85 - Which one is DB?
Answer: Thanks. Populus trichocarpa genome is Phytozome v13.
Figure 2 appears before the citation in the text!
Answer: Thanks. We have moved Figure 2 after the citation in the text.
Figures 2, and 6, part of Figures 4, 5, and 7, are poor quality.
Answer: Thank you for your helpful suggestion. We have improved the quality of these Figures.
We ourselves have done additional changes:
We have revised “PtrFLAs” with “PtrFLA genes” in line 95.
We have revised “eucalyptus” with “Eucalyptus”in line100, 306.
We have revised “poplar” with “P. trichocarpa” in line 112.
We have revised “PtrFLA8 and 31” with “PtrFLA8 and PtrFLA31” in line 119.
We have revised “PtrFLA20 and 38” with “PtrFLA20 and PtrFLA38” in line 120.
We have revised “AtFLA6, 9 and 13” with “AtFLA6, 9, 13” in line 120.
We have revised “PtrFLA5 and 26” with “PtrFLA5 and PtrFLA26” in line 121.
We have revised “the possible similar functions with AtFLA12” with “that they may have similar functions” in line 125.
We have deleted “PtrFLA” in line 127.
We have revised “sub-clade” with “sub-clade I” in line 127.
We have revised “group A PtrFLA member” with “PtrFLAs of group A” in line 157.